# Application of Data Fusion in Traditional Chinese Medicine: A Review

**DOI:** 10.3390/s24010106

**Published:** 2023-12-25

**Authors:** Rui Huang, Shuangcheng Ma, Shengyun Dai, Jian Zheng

**Affiliations:** 1National Institutes for Food and Drug Control, Beijing 102629, China; huangrui2019513@163.com (R.H.); masc@nifdc.org.cn (S.M.); 2School of Traditional Chinese Pharmacy, China Pharmaceutical University, Nanjing 211198, China

**Keywords:** data fusion, traditional Chinese medicine, chemometrics, quality control, origin traceability

## Abstract

Traditional Chinese medicine is characterized by numerous chemical constituents, complex components, and unpredictable interactions among constituents. Therefore, a single analytical technique is usually unable to obtain comprehensive chemical information. Data fusion is an information processing technology that can improve the accuracy of test results by fusing data from multiple devices, which has a broad application prospect by utilizing chemometrics methods, adopting low-level, mid-level, and high-level data fusion techniques, and establishing final classification or prediction models. This paper summarizes the current status of the application of data fusion strategies based on spectroscopy, mass spectrometry, chromatography, and sensor technologies in traditional Chinese medicine (TCM) in light of the latest research progress of data fusion technology at home and abroad. It also gives an outlook on the development of data fusion technology in TCM analysis to provide references for the research and development of TCM.

## 1. Introduction

Traditional Chinese medicine (TCM) is medicine whose clinical application is guided by traditional Chinese medical theories. According to the former study, which showed that TCM contributed to the treatment of COVID-19 owing to its efficacy and comprehensive therapeutic theory, this resulted in its widespread use worldwide [1]. TCM has a long history of use and is mainly derived from natural medicines and their processed products, which have comprehensive therapeutic effects on a variety of diseases, reliable clinical efficacy, and fewer toxic side effects. However, TCM is a complex system, characterized by a large number of chemical compositions, unclear pharmacological mechanisms of action, and uneven quality [2]. Additionally, the traditional evaluation of TCM relies on the human senses of taste, vision, smell, and touch, which are highly subjective and unstable. With the modernization and internationalization of TCM, more and more modern techniques have been used to comprehensively evaluate the quality of TCM, to elucidate the group of chemical substances exerting medicinal effects, and to produce TCM preparations, such as spectroscopy, chromatography, mass spectrometry, and sensor technology. After a long period of exploration and practice, a relatively complete system of analysis of TCM has been formed. It is difficult for a single analytical technique to measure the component information to fully reflect the information of TCM and give a comprehensive evaluation, which can be effectively solved by fusing the information with data [3]. Multiple instrumental analytical data often provide complementary information, and data fusion strategies can combine these data with chemometric methods to obtain more comprehensive information and better classification and prediction results [4]. This paper summarizes the application of data fusion strategies in TCM by citing the literature published from 2010 to 2023 and providing an outlook on its future development.

## 2. Data Fusion Technology

### 2.1. Introduction to Data Fusion Technology

Data fusion techniques are methods of joining multiple data sources to obtain more accurate and comprehensive information. Its advantage is that the data can be separated from the original analysis techniques and other independent data to build a new model and use it for the calculation and speculation of the results [5], which can be applied qualitatively and quantitatively. As a technology with more accurate judgment and higher precision, data fusion is commonly used in automation [6], the mining industry [7], aerospace [8,9,10], military technology [11], agriculture [12], and medical science [13], etc. In the face of high-precision testing needs, data fusion always achieves good results. Ji et al. [14] used fluorescence lifetime imaging microscopy (FLIM) to image exfoliated cervical cells and analyzed the images through an unsupervised machine learning method to build a model for predicting the risk of cervical cancer. The sensitivity and specificity of the model were 91% and 100%, respectively, which were significantly higher than that of the traditional cytological methods, demonstrating that data fusion technology can be used as an effective means of cervical cancer screening and detection. In recent years, the application of data fusion technology has gradually become widespread. In addition to its application in scientific and technological disciplines, it has also gradually begun to be applied to the food industry. Obisesan et al. [15] used HPLC equipped with two detectors, ultraviolet (UV) and charged aerosol (CAD), to identify the origin of palm oil, fused the data measured by the two detectors, and the fusion was able to better validate the geographic origin of the palm oil compared to the results of modeling the classification established by a single technique, which proves the feasibility of the data fusion technology.

### 2.2. Data Fusion Strategy

Based on the data fusion structure the fusion strategies can be categorized into Low-level, Mid-level, and High-level data fusion [16], schemes of which are shown in Figure 1.

In low-level data fusion, also known as data-level fusion, all data are simply stitched together in a sample series into a single matrix whose rows are the number of samples and columns are the number of variables measured by different techniques [17]. Because of its characteristics of large sample size, computational inefficiency, and inability to perform direct fusion for data with different magnitudes, low-level fusion is subject to pre-processing and variable screening, which usually requires pre-processing of the data such as normalization, noise reduction, and deletion of redundant variables before fusion [18]. Low-level fusion is easier to operate than mid- and high-level fusion and can obtain comprehensive information. It is simple to operate, has high detection accuracy, and is suitable for samples with small amounts of data information, e.g., low-level data fusion can better recognize different types of fruit juices [19], identify the authenticity of olive oil as well as origin traceability [20,21].

Mid-level fusion, also known as feature-level fusion, first extracts feature information from each data source, and then analyzes and processes the feature information of each part in tandem, usually using principal component analysis (PCA) or partial least squares discriminant analysis (PLS-DA) [22]. The advantages are that it reduces the amount of data, eliminates irrelevant variables, and improves the computational efficiency, so in most cases, the fusion effect is better than the low-level data fusion [23], which is also the most used fusion method in data fusion strategies. Mid-level fusion feature information should be considered when dealing with data information of high complexity to reduce variables and shorten the calculation time. The mid-level fusion technique is also commonly used in the quality evaluation and origin traceability of food products [24,25], where spectroscopy and mass spectrometry data are fused for analysis [26].

High-level data fusion, also known as decision-level fusion, establishes classification and prediction models for data from different sources separately, and then integrates the results of multiple analyses to obtain the results of joint decision-making. The advantages are strong anti-interference and good fault tolerance; the disadvantages are high processing cost, cumbersome operation, and loss of a large amount of detailed information [25]. High-level data fusion usually includes low-level fusion and mid-level fusion of data, and the key to fusion is to choose the appropriate model. The commonly used models include the Bayesian model, voting mechanism [27], and fuzzy set theory [28]. Li et al. [29] used mid-infrared (MIR) and Raman spectroscopy to determine the content of high fructose syrup (HFGS) in perch honey mixtures and established a partial least squares (PLS) model to predict the concentration of adulterants and, at the same time, adopted three data fusion strategies, low, mid, and high, to improve the prediction accuracy of the quantitative model. The complementary advantages of MIR and Raman spectroscopy indicate that the high-level fusion strategy can be used as a reliable tool for quantitative analysis. The construction of high-level data fusion is more complicated, but the benefit lies in its separate treatment of data blocks, which reduces the mutual interference between different models. High-level fusion strategies can solve the problem of low accuracy of low-level and mid-level fusion and are suitable for analysis scenarios that require higher accuracy. They are commonly used to solve classification problems, such as the classification of hyperspectral images [30], the classification of wild and cultivated mushrooms [28], and the classification of wine varieties [31].

Data fusion techniques can combine information from multiple sources to obtain more comprehensive data through complementary information. The results of research in several fields have shown that models built using data fusion strategies embody more accurate detection and utility than models built with data from a single source [32,33,34,35].

### 2.3. Pre-Processing

Data pre-processing strategy is the process of preparing raw data and making it suitable for machine learning models. For better multivariate analysis, data from a single source needs to be preprocessed to properly scale the data, to reduce noise, and remove uninformative systematic variations. Common data pre-processing includes standard normal variate (SNV), multiplicative scatter correction (MSC), Savitzky-Golay (SG), Wavelet Denoising (WDS), baseline, normalization, and noise, etc. MSC eliminates scattering due to uneven particle distribution and size. The derivative algorithm eliminates interference from baseline drift or background, separates overlapping peaks, and improves resolution and sensitivity [36]. Smoothing can be used to reduce random errors [37]. Indeed, different data sources require special treatment according to characteristics. Spectrum pre-processing is an essential component in the near-infrared (NIR) calibration. SG [38] and multiplicative MSC are worth testing first, likewise, standard normal SNV was used for mid-infrared (MIR) data [39]. Meanwhile, baseline shifts in infrared spectroscopy can be eliminated with derivatives [40,41]. UV spectroscopy often uses baseline corrections [42], while mass spectrometry uses normalization [41]. In addition, due to the complex sample matrix, scaling of the fused matrix is often used as a data pre-processing method to construct suitable models [42].

## 3. Traditional Chinese Medicine and Data Analysis

### 3.1. Problems and Challenges Facing Traditional Chinese Medicine

TCM has a long history of being used for the prevention and treatment of diseases, and as early as thousands of years ago, there was a work, Shennong Ben Cao Jing (Classic of the Materia Medica of the Divine Husbandman), which systematically described the theories, formulas, and functions of TCM [43]. With people’s emphasis on health and the booming development of TCM-related industries, TCM has become a part of the global healthcare system [44], however, with the increasing demand for TCM, more and more problems have been exposed. TCM came from nature: the composition is complex, the mechanism of action is not clear, the safety and efficacy evaluation system is not yet perfect, and the evaluation data are not enough to be recognized by the modern regulatory agencies in Western countries [45]. In the production of TCM preparations, the quality cannot be controlled online, because changes in parameters in the real-time process are not detected in time, and the process parameters set in advance are only applied for the application.

Regardless of the system of medicine, consistent quality of drugs is a prerequisite for stable efficacy, so there is an urgent need for a means to accurately test and analyze the application of TCM in various fields.

### 3.2. Data Analysis Techniques and Traditional Chinese Medicine

With the rapid development of modern analytical and computer technologies, different analytical strategies and methods have also been widely adopted in TCM. Equally, some means of modern data analysis techniques have also been used in the field of TCM, such as chemometrics [46,47], fingerprinting [48,49], metabolomics [50], machine learning [46,51,52], and deep learning [53,54,55].

Chemometrics is an emerging interdisciplinary discipline based on computers and mathematics, which establishes a link between the measured value of a chemical system and the state of the system through statistical or mathematical methods, and has been widely used in the identification [56], origin traceability [57], qualitative characterization [58], quality evaluation [59] and other research in TCM, as well as the entire process of TCM production, including the design of the preparation process and optimization [60], online monitoring [61], and so on. Pattern recognition is an important part of chemometrics, including traditional recognition modes (HCA, PCA, PLS-DA, etc.) and emerging recognition modes (SVM, ANNs, etc.), usually used with fingerprinting, metabolomics, and data fusion.

The technique of chromatography fingerprinting of TCM has been proven to be a comprehensive strategy for evaluating the complete quality of TCM, which can comprehensively reflect the types and quantities of chemical components contained in TCM and its preparations, as well as describe and evaluate the quality of medicines as a whole [50], which is in line with the holistic theory of TCM. Han et al. [62], to better control the quality of Flos Puerariae (saffron), used chemical fingerprinting and chemometric methods for qualitative and quantitative analysis. First, High Performance Liquid Chromatography (HPLC) was used to obtain its fingerprint, followed by a screening of chemical markers by hierarchical cluster analysis (HCA), principal component analysis (PCA), and orthogonal partial least squares discriminant analysis (OPLS-DA). Subsequently, the chemical constituents in Flos Puerariae (saffron) were identified using HPLC with HPLC-FT-ICR-MS. Finally, HPLC was used to quantify the characterized components in Flos Puerariae. The results showed that six peaks in the fingerprint profile were considered as qualitative markers, and their contents were determined separately. In conclusion, fingerprint profiling combined with chemometric methods can discover chemical markers, and on this basis, if further spectroscopic studies are carried out, the quality of TCM can be combined with its therapeutic efficacy, which can help to elucidate the mechanism of action of TCM.

Metabolomics, along with genomics, transcriptomics, and proteomics, forms the basis of systems biology, and in conjunction with chemometrics techniques, can generate high-quality data and identify potential biomarkers [47]. Metabolomics aims to provide a comprehensive metabolomic characterization of system-wide metabolic changes induced by holistic environmental interventions, as well as a comprehensive metabolomic characterization of complex biological systems [63], which is consistent with the holistic view of TCM and therefore is widely used in assessing the quality of TCM, clinical efficacy, and the essence of TCM syndrome. Yi et al. [64] combined GC-MS with chemometrics to analyze the metabolomics of Chenpi and Qingpi. Firstly, a total of 24 samples of Chenpi at the first fruiting stage and fully ripened stage were analyzed for volatile constituents by gas chromatography-mass spectrometry (GC-MS), and the problem of overlapping some of the peaks was resolved using alternatively moving window factor analysis (AMWFA). Finally, the qualitative and quantitative results of 82 volatile components were also obtained. In addition, the metabolic footprints during fruit ripening were revealed with the help of PCA, thermogram analysis, and Pearson’s correlation analysis, and the characteristic compounds representing three different stages of ripening were screened out. Therefore, the combination of metabolomics analysis and chemometrics will be an effective strategy for the quality control of proximate TCM.

The deep learning method is one of the machine learning methods that can actively screen the data and extract high-dimensional features of the data. Additionally, the prediction and classification accuracy of the data is higher than traditional machine learning [61], Kabir et al. [47] investigated hyperspectral imaging combined with the CNN network to recognize the Fritillaria thunbergii varieties, and the performance of the two models, PLS-DA and SVM, was compared with CNN. The results showed that CNN had the highest accuracy on both training and test sets, 98.88% and 88.89%, respectively, followed by 92.59% and 81.94% for PLS-DA and 99.65% and 79.17% for SVM. This study demonstrated that hyperspectral imaging combined with deep learning can quickly, nondestructively, and accurately identify Fritillaria thunbergii varieties. However, multi-source information often contains a lot of redundant information, which can be feature extracted using a data fusion strategy; therefore, combining data fusion strategies from multi-source instruments and deep learning for traditional Chinese medicine is a new direction and technique for future research.

Data analysis technology has mined more useful information for TCM through analyzing TCM from various aspects and angles. Data fusion technology is also one of the data analyses that can provide more accurate sample information and better inference. Additionally, it has become gradually better known and widely used in the field of TCM in recent years due to its high accuracy and complementary information.

### 3.3. Model Validation

Data fusion is a powerful approach which provides more comprehensive information. However, as the sample information continues to increase, the model complexity and untrained parameters will continue to expand. At this point, the model is highly susceptible to overfitting, which is characterized by a model that has less loss and higher prediction accuracy on the training data, but the model’s loss on the test set will be large and less accurate. This happens in many studies, so proper validation techniques for these models are essential to avoid overfitting. Commonly used validation methods are internal cross-validation or external dataset testing [65].

What the cross-validation method does is try to do multiple different sets of training/testing of the model using different training/testing set divisions to deal with the problem of overly one-sided word test results and insufficient training data. The training set is the dataset used to train the parameters within the model, and the Classifier adjusts itself directly to the training set to get better classification results. The validation set is used to check the state of the model during the training process. After the performance of the validation set is stabilized, if the training is continued, the performance of the training set will continue to rise, but the validation set will fall instead of rising: overfitting generally occurs, so the validation set is also used to determine when to stop training. The external test set is used to evaluate the model generalization ability, and thus determine whether this model works or not. Around 77% of the research articles used external datasets to test the performance of the developed models, while the rest relied on cross-validation procedures. Testing the predictive ability of a model on external datasets is crucial for performance evaluation [66].

Since multivariate analysis is scale-dependent, data from single techniques are usually preprocessed to properly scale the data. It is also worth noting that special attention must be paid during validation to justify the appropriate selection of features such as the number of PCs to avoid the over-/under- fitting of the models [66]. If the model is too simple with few parameters, the deviation between the predicted values and the correct values we try to predict will be high, resulting in an underfitting phenomenon. On the contrary, if the model has a large number of parameters to respond to all the information from the data source, an overfitting phenomenon occurs that performs well on the training set but has a high error rate on the test set [67]. This remains a problem for all datasets: that is, the balance between data overfitting and underfitting. So, when using data fusion techniques, it is important to pay special attention to the problem of overfitting and underfitting of the model after fusion.

## 4. Application of Data Fusion Technology in Traditional Chinese Medicine

### 4.1. Data Fusion for Different Spectroscopy Data

Spectroscopy is one of the most widely used analytical instruments in TCM. In general, spectroscopy data is a matrix of M × N, in which M represents a row of samples and N represents a column of variables. The matrix of M × N is the input variable, while the output variable can be many, such as content, classification status, etc. There are common spectroscopic techniques such as ultraviolet-visible spectroscopy (UV-VIS), Raman Spectroscopy, near-infrared spectroscopy (NIR), and mid-infrared spectroscopy (MIR), etc. The main advantages of spectroscopy are simple pre-processing, simple analysis, low cost, and the ability to perform real-time control. Data fusion between different kinds of spectra is widely used in TCM. Based on the complementary nature of different spectral information, the fusion strategy can obtain more comprehensive data.

#### 4.1.1. Data Fusion for Infrared Spectroscopy Data

The most common is the data fusion between MIR and NIR, which can quickly and accurately identify the sample information as well as determine the species and content of organic matter based on the sample’s characteristic peaks and specific absorption spectroscopy [68]. Fu et al. [69] used NIR and MIR combined with chemometrics to identify Angelica dahurica from different origins, established a supervised pattern based on the PLS-DA algorithm, and were able to accurately identify the geographic origin of Angelica dahurica. Pei et al. [70] use FT-MIR and NIR spectroscopy combined with low, mid, and high-level data fusion, based on software such as OMNIC 9.7.7, SIMCA-P+13.0, MATLAB R2017a, and other software to carry out data processing and modeling analysis, which successfully realized the origin traceability of 196 species of Wild Paris polyphylla var. yunnanensis through the establishment of PLS-DA and a random forest (RF) classification model, in which the accuracy of the model origin identification established by the high-level data fusion of the extraction of the principal component feature variables was 100%. This proves the feasibility of the 2 types of spectroscopy fusion for the accurate origin traceability of Wild Paris polyphylla var. yunnanensis, and the flow of data fusion is shown in Figure 2. To ensure the safety of the clinical medication, Sun et al. [71] used NIR and infrared spectroscopy (IR) to differentiate between official and unofficial rhubarb, and they compared four classification methods, namely PLS-DA, SIMCA, SVM, and ANN, and the results showed that fusion using the ANN model resulted in 100% classification accuracy. Qi et al. [72] assessed the quality of Rhizoma Coptidis from different origins based on quantitative and qualitative metabolic spectroscopy obtained by HPLC in combination with FT-NIR and FT-MIR. First, eight alkaloids were determined by HPLC to screen out the index compounds that could be used as indicator compounds for the quality identification of Huanglian, and then a single-technique model and a spectroscopy fusion model were established concerning the content data of the index compounds. The results showed that the fusion of spectroscopy data with the partial least squares regression (PLSR) algorithm could effectively determine the content of the index compounds and achieve better prediction results, and the fusion technique could provide a more comprehensive and effective quality assessment of Rhizoma Coptidis. Zhang et al. [73] studied the extraction process of Xiao’er Xiaoji Zhike Oral Liquid (XXZOL) by first determining the concentrations of seven key quality attributes during the extraction process, and then establishing low-level and mid-level fusion models based on two types of spectroscopy data in the NIR and MIR with PLS. The results of MATLAB 2019a software showed that the prediction of the data fusion model was better than that of the spectroscopy model for seven key quality attributes, which led to faster, more comprehensive, and more complete detection of information in the extraction process. Hai et al. [74] collected spectroscopy information of 320 Dendrobium huoshanense (DHS) using nano effects near and mid-infrared, extracted their eigenvariables, and performed mid-level data fusion. The resulting eigenvector projection (VIP) was combined with PLS-DA and orthogonal partial least squares discriminant analysis OPLS-DA to determine the age of the DHS with a discrimination accuracy of 100%. This study demonstrated that the method of multi-spectroscopy eigenvector fusion is a reliable analytical method to accurately identify the cultivation age of TCM. Wang et al. [75] employed a low, mid, and high-level fusion strategy using FT-NIR and attenuated total reflectance Fourier transform mid-infrared spectroscopy (ATR-FT-MIR, ATR-FT-MIR) combined with chemometrics for the TCM Eucommia ulmoides leaves from 13 provinces in China to geographic traceability. The results showed that the identification results of the high-level data fusion strategy were better than those of the low and mid-level fusion strategies, probably because the high-level fusion which occurred at the decision level and could weaken the influence of irrelevant or interfering information. Li et al. [76] identified the geographic sources of Panax notoginseng from five regions in Yunnan Province using low-level, mid and high-level fusion strategies by using FT-MIR and NIR junction RF algorithms, and then fused the results of the decisions using SVMs. The results showed that the mid and high-level data fusion strategies provided better identification results than the low-level fusion strategy, probably because the low-level fusion contained too much useless raw information. Similarly, Yang et al. [77] constructed a classification model for Panax notoginseng with different mixing ratios by using analysis software such as Unscrambler 10.4, OriginPro 9.0, and MATLAB 2018a to perform low, mid, and high-level data fusion using NIR and MIR of Panax notoginseng doped samples of different grades. The results showed that the application of the data fusion technique successfully improved the identification of adulterants in Panax notoginseng powder compared with the results of the single-spectrum analysis.

#### 4.1.2. Data Fusion for Other Spectroscopy Data

Li et al. [78] geographically traced Atractylodes macrocephala Koidz to the low-level and mid-level fusion strategy based on chemometrics classification combined with fluorescence and Raman spectroscopy techniques and established principal component analysis-linear discriminant analysis (PCA-LDA), PLS-DA and RF classification. The results showed that the correct classification rate of PLS-DA based on low-level data fusion could reach 91.0%. Wang et al. [79] used laser-induced breakdown spectroscopy (LIBS) and IR combined with RF to identify Radix Astragali from different geographic regions based on low-level fusion, and unsupervised discriminative model using PCA to identify Radix Astragali, the results showed that three Radix Astragali samples could not be accurately differentiated by PCA, and the mid-level data fusion was based on the low-level fusion data, the feature variables were extracted by variable importance (VI) metrics, and the feature values were used as input variables to construct RF discriminant models. The results showed that the RF model of mid-level data fusion had the best prediction performance with an accuracy of 97.78%. Jiang et al. [80] used NIR and ultraviolet spectroscopy (UV) combined with PLSR to establish low-level data fusion and mid-level data fusion models for the simultaneous determination of six ginsenosides and four saccharides in Shenmai injections, and the results showed that the data fusion strategy under the synergistic effect of the complementary information of the two spectroscopy improved the efficiency of the quantitative analysis of ginseng saponins and saccharides in a better way compared to the modeling established by independent NIR or UV spectroscopy.

These studies have demonstrated that data fusion between spectroscopy can improve the accuracy of predictive models, and the method of fusing data from different spectroscopy has been very common in the field of TCM, especially the fusion between NIR and MIR data. IR spectroscopy can also be fused with inductively coupled plasma-atomic emission spectroscopy (ICP-AES) for the determination of trace and mineral elements in TCM [81]. It has been proved that integrating and combining spectroscopy and chemometrics is very feasible in the field of TCM.

### 4.2. Data Fusion for Spectroscopy Data and Other Techniques Data

Different types of origin traceability techniques have their own advantages, and after fusion, complementary information can be obtained from various aspects. Therefore, in addition to the fusion of different spectral data, spectra can also be fused with mass spectrometry and chromatography techniques. Thus, in addition to the mutual fusion of different spectral data, spectra can also be fused with mass spectrometry and chromatography. Table 1 lists some applications of such fusion in recent years.

Commonly, there is the fusion of chromatography and spectroscopy. Chromatography has a good separation function and can analyze the sample quantitatively, while spectroscopy can suggest the different chemical groups contained in the sample and can perform qualitative analysis. Combining the two can complement each other’s information and provide a comprehensive analysis of TCM with complex compositions. Yan et al. [82] used a combination of UV spectroscopy fingerprinting and HPLC fingerprinting combined with antioxidant activity determination and analyzed the results of the data based on the fingerprint super-informative feature digital evaluation system and SIMCA software to qualitatively and quantitatively evaluate the quality consistency of 26 batches of Liuwei Dihuang Pill, respectively. The two types of fingerprints reflected the quality of the drug from different perspectives, and the complementary information provided a rapid, effective, and accurate method for the evaluation of Chinese medicinal preparations. Yu et al. [83] used the fusion of spectroscopy and chromatography data to identify swertia leducii and its closely relatives, collected Fourier Transform Infrared Spectroscopy (FTIR) data and HPLC fingerprinting data of 102 samples and preprocessed them accordingly. Then low-level and mid-level fusion strategies were used to establish a RF discrimination model of spectroscopy and chromatography fingerprinting data, and the results showed that mid-level fusion of the data was most effective, and all the samples were correctly categorized, which demonstrated that the model constructed was able to discriminate the different genera of swertia leducii well. Wu et al. [84] used ultra-performance liquid chromatography (UPLC) to determine the content of components on a heavy floor, fused the results with the data collected by FTIR spectroscopy at low and mid-level levels, and simultaneously established PLS-DA, SVM, and RF Recognition mode models for the origin traceability of heavy floor plants of the Paris species. The results showed that the mid-level fusion model accuracy of PLS-DA was superior to that of the low data fusion model and the other models, which proved that the mid-level data fusion in conjunction with chemometrics methods could correctly identify different Paris species and could trace their geographic origins.

Mass spectrometry can determine the molecular mass and structural formula, which has the characteristics of rapidity, high sensitivity, and a wider range of use [85], the combination of mass spectrometry data with spectroscopy data can expand the scope of analysis, analyzing the complex composition of TCM compound formulas, and obtain the information of unknown TCM. Dai et al. [86] fumigated four geographically different sources of Ophiopogon japonicus Ker-Gawler (Liliaceae) and analyzed them metabolically by using NIR and UHPLC-LTQ-Orbitrap mass spectrometry. The effect of data fusion on the spatial distribution of the samples was investigated by fusing the NIR spectroscopy data with the information of the samples collected by UHPLC-HRMS. Firstly, based on the PCA, the discriminatory power of each technique was examined separately, and secondly, the NIR and UHPLC-HRMS datasets were compared for their classification results using mid-level fusion without variable selection and mid-level fusion with variable selection, respectively. The results showed that the model discriminatory power was significantly improved after data fusion. Meanwhile, Song et al. [87] used a data fusion technique to predict the antioxidant capacity and total phenolic content of bearberry leaves. Firstly, the spectroscopy fingerprints of bearberry leaves were obtained using UV-visible spectroscopy, and secondly, the metabolomics analyses were performed using ultra-high-performance liquid chromatography with time-of-flight mass spectrometry (UHPLC-Q-TOF-MS), and mid-level fusion of spectroscopy and mass spectrometry data was performed by using PLSR model. The results showed that the RPDs value and DPPH of this method were 6.258 and 6.699, respectively, which proved that the method had the excellent predictive ability.

**Table 1 sensors-24-00106-t001:** Applications of spectral and other data fusion.

TCM	Analytical Techniques	Chemometrics	Fusion Level	Ref.
Gentiana rigescens	IR, UV	PLS-DA, SVM	mid	[88]
Leccinum rugosiceps	IR, UV	PLS-DA, SVM	mid	[89]
Chinese herbal injection	NIRS, UVS	PLS, UVPLS	low, mid	[80]
cultivated Macrohyporia cocos	ATR-FTIR, UFLC	PLS-DA, PLSR	-	[90]
Wolfiporia cocos	FT-NIR, FT-MIR	PLS-DA, SVM PCA, HCA	low, mid	[91]
wildParis polyphylla Smith var. yunnanensis	FT-MIR, UV-Vis	SVM-GS, RF	low, mid, high	[92]
Polygonatum kingianum	ATR-FTIR, UV-Vis	PLS-DA, PCA, HCA	Low, mid, high	[93]
Dendrobium huoshanense	nano-effect NIR nano-effect MIR	PLS-DA, OPLS-DA	mid	[75]
Radix puerariae	NIR, UV	PLSR	low	[94]
Lonicera japonica and Artemisia annua	NIR, MIR	C-PLS, SO-PLS	-	[95]
Dendrobium Species	NIR, UV	SVM, PLS-DA	low, mid, high	[96]
Coptidis Rhizoma	NIR, MIR	PLS-DA, PLSR	low, mid, high	[97]
Radix Astragali	Vis-NIR, NIR	PCA, CNN, SVM, LR	-	[98]
Boletus bainiugan	FT-NIR, FT-MIR	PLS-DA, SVM 3DCOS-ResNet	low, mid	[99]
Honey	Gas sensors, Liquid sensors	LDA	low	[100]
Macrohyporia cocos	FTIR, LC	PLS-DA	low, high	[101]
Macrohyporia cocos	FTIR, HPLC	PLS-DA, PCA	mid	[102]
Swertia leducii	FTIR, UPLC	HCA, RF	low, mid	[83]
sulfur-Ophiopogonis Radix	NIR, UHPLC-HRMS	PCA, PLS-LDA	mid	[86]
Curcumae Rhizoma	FT-NIR, E-nose, colormeter	PLS-DA	mid	[103]

### 4.3. Data Fusion for Mass Spectrometry and Chromatography

#### 4.3.1. Data Fusion for Mass Spectrometry Data

The integration of mass spectrometry in TCM has fewer applications, probably because mass spectrometry is seldom used independently, and most of them are used in conjunction with chromatography, such as gas chromatography-mass spectrometry (GC-MS), liquid chromatography-mass spectrometry (LC-MS), and so on. Massaro et al. [104] used real-time high-resolution mass spectrometry (DART-HRMS) with two modes of positive and negative ions to collect data on genuine and counterfeit Oregano, and mid-level data fusion coupled with supervised chemometric modeling was used to achieve discrimination between genuine and counterfeit Oregano, with a sensitivity and specificity of more than 90%. Zhang et al. [105] used headspace gas chromatography-mass spectrometry (HSGC-MS) and ultra-high performance liquid chromatography-quadrupole time-of-flight mass spectrometry (UHPLC-QTOF/MS) techniques to analyze metabolites of five chrysanthemum flower near-relatives; HSGC-MS was used for volatile metabolites, and as a complementary approach, UHPLC-MS was used to efficiently characterize the nonvolatile, and the fusion of the data from the two mass spectrometry and spectroscopy provided comprehensive and complementary information for the chrysanthemum flower metabolomics study.

#### 4.3.2. Data Fusion for Chromatography Data

Chromatography is the only common analytical instrument with a separation function, which can qualitatively and quantitatively analyze the separated components according to different chemical or physical properties. It has the advantages of high selectivity, high sensitivity, and high separation capacity, which is suitable for the identification of compounds, and when equipped with different detectors can obtain different analysis results [106]. Chromatography fingerprinting is also commonly used in TCM to obtain characteristic chromatograms of compounds, and the fusion of fingerprint data at different wavelengths allows for the creation of a new chromatogram as a way to retain more chemical information [107]. Shen et al. [108] traced the origin of the herbs according to the growing environment and latitude and longitude of Gentiana Rigescens, using HPLC combined with a diode array detector (DAD) to generate chromatography fingerprints of the aboveground and underground parts of Gentiana Rigescens, and used two supervised identification techniques, RF and OPLS-DA, to establish a classification model, which showed that the classification effect of OPLS-DA is better than that of RF. The accuracy of the classification model constructed by utilizing the low-level data fusion method is more than 95%, which has better recognition and prediction ability.

Currently, there are relatively few studies on the fusion of mass spectrometry and chromatography data in TCM, because when the intensity of the chromatography peaks is not of the same magnitude, different degrees of quality deviation will occur, and the contamination of the mass spectrometry instrument can lead to intensity drift. The change or aging of the chromatography column can lead to the drift of retention time, which makes the raw signal produced by the instrument not stable enough to maintain the model’s quality, so there is a need to develop some data pre-processing and other mathematical methods [109,110]. However, mass spectrometry can determine the molecular mass and structural formulae, and chromatography has high separation efficiency and fast analytical speed. Therefore, the fusion of the two is very effective and has been widely used in food [111,112] and other fields.

### 4.4. Data Fusion for Sensor Data

The traditional evaluation of TCM mainly relies on human taste, vision, smell, and touch, yet this reliance on experience is highly subjective and poorly reproducible. Sensor technology, as a newly developed detection device, obtains information quickly and accurately, and the fusion of data from different sensors can make up for the inadequacy of information from a single sensor [113]. It has been widely used in the fields of computers [114], agriculture [115], transportation [116], medicine [117], and so on. With the development of technology, in addition to the mutual fusion between sensor technologies, there is also the fusion of sensor technology with other technologies, which has been widely used in the identification of varieties, source identification, and quality evaluation for TCM [118,119]. Zhang et al. [120] used electronic nose (e-nose) and electronic tongue (e-tongue) data combined with chemometrics to qualitatively identify and quantify the quality of Xiaochaihu granules (XCHG). Firstly, the main chemical components such as saikosaponin b2, baicalin, and glycyrrhizin were quantified by UHPLC. Secondly, the data of odor, color, and taste of XCHG were measured by the e-nose, e-eye, and e-tongue. Finally, the sensor data were fused with the chromatography data, and the predictive model for the content constituents was established using PLSR. Wang et al. [121] used two sensor devices, an e-nose, and an e-tongue, to establish a discriminative model of Codonopsis Radix, selecting the optimal model from all combinations of models trained by the two pre-processing methods and the three classification methods, and the results showed that the PLS-DA could well discriminate the original plant source of Codonopsis Radix after modeling. Miao et al. [122] proposed a method for the fusion of e-nose and NIR data combined with SVM used for the identification of different species of ginsengs. Six commonly used features were extracted from the e-nose, and then the feature data of NIR were extracted by PCA. Finally, the models were constructed and trained using a support vector machine (SVM). The results showed that the high-level data fusion method combined with SVM achieved a good classification performance of 99.24%. Jing et al. [123] used an e-nose and e-tongue to measure the characteristic odors of two different species of Magnolia Officinalis Cortex, used UHPLC to quantify eight active components in Magnolia Officinalis Cortex, used Origin Pro2021b software to analyze the results with chemometrics, such as PCA, HCA, PLS-DA, LDA, etc., and used the low and mid-level data fusion to establish a classification model, respectively, in which the model discrimination accuracy of the mid-level fusion strategy reached 100%. The results demonstrated that this method is a powerful tool to differentiate between the two types of Magnolia Officinalis Cortex sources.

## 5. Comprehensive Evaluation of Data Fusion

In summary, data fusion technology has obvious advantages when facing the needs of multi-sample and high-precision detection, and relying on the complementarity as well as synergy of information, data fusion technology can better respond to the overall information of TCM, which is in line with the “holistic view” of TCM. However, there are some limitations in the choice of fusion strategy. 

Data fusion is not the pursuit of high-level, high-level data fusion can meet the requirements of high-precision detection, but its operation is cumbersome, and it may not necessarily achieve better results than low-level or mid-level fusion [101]. Low-level data fusion is simple to operate, more rapid and convenient than mid-level and high-level data fusion techniques, and its advantage lies in the complementary nature of various types of data, which is suitable for samples with a small amount of information data and low detection requirements. However, it contains too much redundancy, interference, noise, and irrelevant information, and when the amount of information data is huge, the low-level data fusion consumes a lot of time, and it is difficult to meet the detection requirements. At this time, it is necessary to consider the application of the mid-level or high-level data fusion strategy. For data information with high complexity, mid-level data fusion should be used to find out the feature information in the data to reduce the variables and reduce the computation time, mid-level data fusion eliminates irrelevant variables and improves the computational efficiency, so this strategy is one of the most used methods in data fusion [124], but we should pay attention to the problem of the decrease in accuracy after mid-level fusion. The high-level fusion method has high accuracy and can solve the problem of accuracy degradation due to the reduction of variables in mid-level fusion, but the exploration of its decision layer consumes a lot of time and the model construction is more complicated, so it is only applicable to scenarios with a large sample size and the need for real-time high-precision analysis. The application of different fusion strategies does not have a unified standard, so in practice, data of different natures should be analyzed by multiple levels of fusion strategies, and then filter out the optimal data fusion method.

In terms of model selection, the commonly used models for classification are PLS-DA, SVM, RF, etc., and the commonly used regression models are PCR, PLSR, SVMR, etc., which can be used to select the appropriate model according to the specific research purpose. It is worth mentioning that the latest deep learning algorithms have far exceeded the traditional machine learning algorithms for data prediction and classification accuracy, deep learning does not require us to extract the features, but will automatically screen the data and extract the data high-dimensional features, so with the development of computer technology, data fusion technology combined with deep learning will become a new development trend in the field of TCM.

## 6. Conclusions and Future Outlooks

Obviously, data fusion technology, by the complementary and synergistic characteristics of information, has had an important impact on the authenticity, safety, and effectiveness of TCM, and has great potential for development in the field of TCM. With the improvement of data fusion technology and the development of the discipline of TCM, we should focus on the use of data fusion technology to break through the detection difficulties in TCM, combining the practical and theoretical, such as data fusion technology being used to simulate the growth environment, temperature, humidity, etc. To cultivate the best quality herbs, a suitable model of traceability and categorization of the origin of TCM is established to ensure the authenticity. Monitoring pesticide residues, heavy metals, and other harmful substances is required to ensure the safety and enhance the quality of TCM. In the production of TCM preparations, data fusion technology can be used to collect multi-point data for each production link such as extraction, concentration, and packaging of finished products to find the best process parameters and realize the precise control of the process. In terms of the effectiveness of TCM, data fusion technology can be used to evaluate the biological indexes, as well as quantitative and qualitative testing of the chemical indexes, to reach the safety and effectiveness of the TCM products.

The data fusion technology in the field of TCM is usually the fusion of physical information such as spectroscopy, mass spectroscopy, or sensors. In the future, chemical information or biological information like DNA and gene sequencing can be fused separately or with each other to obtain all the information about TCM in a more comprehensive and precise way. The current data analysis technology is limited to the fusion of two or three data. The fusion of four, five, or even more data can be tried in the future. The reliability of samples and data is also an issue that needs to be focused on, and the samples should have stronger representativeness and sufficient sample size to enhance the applicability and stability of the model and pave the way for subsequent data fusion. In the future, accurate processing of large and complex data sets will need to rely on other analytical techniques with reliability and specificity for adequate and systematic processing. Therefore, it is expected that data pre-processing techniques will have many innovations in a few years.

However, the data fusion technology itself carries a certain degree of error. If the samples are not correctly categorized, then the predictions will be more confusing. There are no clear limits on what kind of analytical content should be subject to data fusion techniques. For some data with small sample sizes and simple analysis tasks, data fusion undoubtedly adds to the difficulty of the workload. In practice, when data fusion techniques are employed, the preprocessing of information data or model selection is often operated and analyzed according to the thinking of the analysts or procedures set by the software, and there is a lack of standardization. So, in the future this should be combined with the existing research to put forward the application of data fusion strategy based on the standards in TCM.

## Figures and Tables

**Figure 1 sensors-24-00106-f001:**
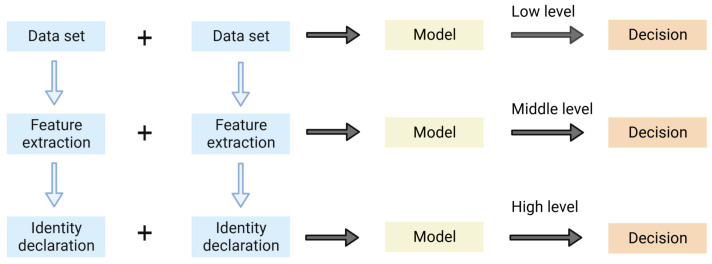
Schemes of the low-, mid-, and high-level data fusion.

**Figure 2 sensors-24-00106-f002:**
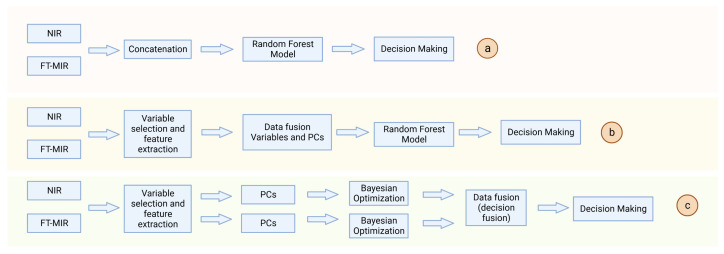
The data fusion process for geographical traceability of Paris Polyphylla Var. Yunnanensis. (**a**) low-level data fusion; (**b**) mid-level data fusion; (**c**) high-level data fusion.

## Data Availability

No data were used for the research described in the article.

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
