# Peer review of "Application of Data Fusion in Traditional Chinese Medicine: A Review"

_sensors, 2023, doi:10.3390/s24010106_

Round 1
Reviewer 1 Report
Comments and Suggestions for Authors
Review of the article "Application of Data Fusion in traditional Chinese medicine: A review."
The paper under review presents an application of data fusion strategies based on spectroscopy, mass spectrometry, chromatography, and sensor technologies in traditional Chinese medicine.
General comments
This topic is exciting and very relevant. However, this article needs improvements.
Firstly, authors should edit the entire article for spelling and formatting.
2.1. Introduction to Data Fusion Technology
"data fusion" is a well-known term that does not need such a long explanation. Section 2.1. must be shortened or modified.
Due to the complex sample matrix, background noise, etc., data processing is essential in data and model preparation. Therefore, the article should expand on data preprocessing, define objectives, and explain data preprocessing strategies.
2.2.
The section requires visualization (schemes of the low-, mid-, and high-level data fusion approaches).
Examples from the literature sources
No. 19 and 25 - do not correspond to the topic of the review article.
4. Application of Data Fusion Technology in Traditional Chinese Medicine
4.1. and 4.2. section contains a huge amount of information. Information must be organized and systematized. Titles for sections 4.1. , 4.2. and 4.3.2. should be renamed to clarify the idea.
4.1
Spectroscopic data matrices, inputs and outputs should be described in the section.
Authors must provide full names for all software.
Line 252 – Please change SIMAC-P to SIMCA-P
4.2.
There is no reference to the Table 1 in the text
Lines 348 and 424 repeat the same idea about chromatography.
Author Response
Comments 1: [Firstly, authors should edit the entire article for spelling and formatting.]
Response 1:Thank you for pointing this out. We have edited the spelling and formatting of the entire article. All corrections have been made in the manuscript in red type.
Comments 2: [2.1. Introduction to Data Fusion Technology "data fusion" is a well-known term that does not need such a long explanation. Section 2.1. must be shortened or modified.]
|
Response 2:Thank you for pointing this out. We agree with this comment. Therefore, we have modified the explanation of data fusion by removing part of the introduction. Mention exactly where in the revised manuscript this change can be found – page 2, Paragraph 2.1, and lines 52-54.
Comments 3: [Due to the complex sample matrix, background noise, etc., data processing is essential in data and model preparation. Therefore, the article should expand on data preprocessing, define objectives, and explain data preprocessing strategies.]
Response 3: Dear reviewer, thank you for pointing this out, we strongly agree with your comment that data processing is essential in data and model preparation, so we have added a new section to introduce data preprocessing and list the preprocessing methods that apply to data from different sources. Mention exactly where in the revised manuscript this change can be found – page 3, paragraph 2.3, and lines 130-146.
|
Comments 4: [2.2. The section requires visualization (schemes of the low-, mid-, and high-level data fusion approaches).]
Response 4: Dear reviewer, we have given the process of low, mid and high-level data fusion schemes in Figure 1, and in Figure 2 we have described in detail the process of geographic traceability of low, mid, and high-level data fusion strategies for Paris Polyphylla Var. Yunnanensis. Mention exactly where in the revised manuscript this change can be found – page 3, paragraph 2.2, and line 127.and page 8, paragraph 4.1.1, and lines 351-353.
Comments 5: [Examples from the literature sources No. 19 and 25 - do not correspond to the topic of the review article.]
Response 5: Agree. The Examples from literature No.19 and 25 were intended to give an example of a low-level, mid-level fusion application to facilitate a better presentation of the two fusion strategies. Still, they do not correspond to the topic of the review article, so we have removed them.
|
Comments 6: [4. Application of Data Fusion Technology in Traditional Chinese Medicine 4.1. and 4.2. section contains a huge amount of information. Information must be organized and systematized. Titles for sections 4.1. , 4.2. and 4.3.2. should be renamed to clarify the idea.]
|
Response 6: Thank you for pointing this out. We agree with this comment. The content of the 4.1 section is mainly based on spectroscopy data, it is also a part of data fusion that is more deeply researched, so there are more references, and we would like to show this part to the reader as much as possible. We summarize the content of this part and divide it into two parts, one part is 4.1.1 about IR spectroscopy, and the other part is 4.1.2 about other spectroscopy. We have changed the title of section 4.2 to “ 4.2 Data fusion for spectroscopy data and other techniques data”. Similarly, we changed the title of section 4.3.2 to “Data fusion for chromatography data”.
|
Comments 7: [4.1 Spectroscopic data matrices, inputs, and outputs should be described in the section.]
Response 7: Thank you for pointing this out. We agree with this comment. We have added a description of the spectroscopic data matrices to section 4.1. Mention exactly where in the revised manuscript this change can be found – page 6, paragraph 4.1, and lines 275-278.
Comments 8: [Authors must provide full names for all software. Line 252 – Please change SIMAC-P to SIMCA-P]
Response 8: Thank you for pointing this out. We agree with this comment. We have provided the full names of all the software, and changed SIMAC-P to SIMCA-P. All corrections have been made in the manuscript in red type.
Comments 9: [4.2There is no reference to the Table 1 in the text Lines 348 and 424 repeat the same idea about chromatography.]
|
Response 9: Thank you for pointing this out. We agree with this comment. Therefore, we have included a reference to Table 1 in the article. Similarly, we have deleted and modified the same description of chromatography in lines 348 and 424 of the original article.
|
Dear reviewer, your valuable comments have been carefully revised, and we sincerely hope that you are in good health and have a good day!

Reviewer 2 Report
Comments and Suggestions for Authors
This review gives a comprehensive review of data fusion techniques with applications to traditional Chinese medicine (TCM) analysis. The review has a good cover of applications and methods used for data fusion. However, one main thing missing is the validation process. Data fusion is a powerful approach but also highly prone to over-fitting. Indeed, many literatures cited by this review actually failed miserably on this aspect. Of course that's not the fault of the authors of this review, they need to remind readers the potential risk of this type of approach. This topic deserves a section in this review and with it in place, this review is ready for publication.
Comments on the Quality of English LanguageThere are many incorrect use of "-" in the manuscript such as "preci-sion", "quali-ta-tive", "tra-ditional" etc. These needs to be corrected.
Author Response
Comments 1: [This review gives a comprehensive review of data fusion techniques with applications to traditional Chinese medicine (TCM) analysis. The review has a good cover of applications and methods used for data fusion. However, one main thing missing is the validation process. Data fusion is a powerful approach but is also highly prone to over-fitting. Indeed, many literatures cited by this review failed miserably in this aspect. Of course, that's not the fault of the authors of this review, they need to remind readers of the potential risk of this type of approach. This topic deserves a section in this review and with it in place, this review is ready for publication.]
|
Response 1: Thank you for pointing this out. We agree with this comment. Many studies have neglected the validation process. We have added a section on model validation, including how to avoid underfitting and overfitting problems, to alert the reader to the potential risks associated with this type of approach as well as suggest measures to mitigate such risks. Mention exactly where in the revised manuscript this change can be found – page 5-6, paragraph 3.3, and lines 238-271.
|
Dear reviewer, your valuable comments have been carefully revised, and we sincerely hope that you are in good health and have a good day! |

Round 2
Reviewer 1 Report
Comments and Suggestions for Authors
I would like to express my gratitude for the effort and dedication that authors put into revising the manuscript.
The manuscript needs some minor revisions:
-) Delete line 445
-) Correct the line 352: Figure.2 to Figure 2.
-) Please increase the resolution of the Figure 2.
Author Response
Comments 1: [Delete line 445.]
Response 1: Dear reviewer, thank you for pointing this out, we have deleted line 445 and we apologize for the additional workload due to our negligence.
Comments 2: [Correct the line 352: Figure.2 to Figure 2.]
Response 2: Dear reviewer, thank you for pointing this out, we have changed “Figure.2” to “Figure 2.” and we apologize for the additional workload due to our negligence.
Comments 3: [Please increase the resolution of the Figure 2.]
Response 3: Dear reviewer, thank you for pointing this out, we have made changes and increased the resolution of Figure 2.
Dear reviewer, thank you for reviewing our article. We wish you a Merry Christmas in advance and hope you have a wonderful day

Reviewer 2 Report
Comments and Suggestions for Authors
The revised manuscript has addressed my concern and it is ready for publication.
Author Response
Comment : [The revised manuscript has addressed my concern and it is ready for publication.]
Response : Dear reviewer, thank you for reviewing our article. We wish you a Merry
Christmas in advance and hope you have a wonderful day!
